# Nutrition and Cardiovascular Health

**DOI:** 10.3390/ijms19123988

**Published:** 2018-12-11

**Authors:** Rosa Casas, Sara Castro-Barquero, Ramon Estruch, Emilio Sacanella

**Affiliations:** 1Department of Internal Medicine, Hospital Clinic, Institut d’Investigació Biomèdica August Pi i Sunyer (IDIBAPS), University of Barcelona, Villarroel, 170, 08036 Barcelona, Spain; rcasas1@clinic.cat (R.C.); sacastro@clinic.cat (S.C.-B.); restruch@clinic.cat (R.E.); 2CIBER 06/03: Fisiopatología de la Obesidad y la Nutrición, Instituto de Salud Carlos III, 28029 Madrid, Spain

**Keywords:** Mediterranean diet, cardiovascular disease, inflammation, nutrients, polyphenols, MUFA, PUFA, bioactive compounds, phytosterols, dietary pattern

## Abstract

Cardiovascular disease (CVD) is the leading cause of death in Western countries, representing almost 30% of all deaths worldwide. Evidence shows the effectiveness of healthy dietary patterns and lifestyles for the prevention of CVD. Furthermore, the rising incidence of CVD over the last 25 years has become a public health priority, especially the prevention of CVD (or cardiovascular events) through lifestyle interventions. Current scientific evidence shows that Western dietary patterns compared to healthier dietary patterns, such as the ‘Mediterranean diet’ (MeDiet), leads to an excessive production of proinflammatory cytokines associated with a reduced synthesis of anti-inflammatory cytokines. In fact, dietary intervention allows better combination of multiple foods and nutrients. Therefore, a healthy dietary pattern shows a greater magnitude of beneficial effects than the potential effects of a single nutrient supplementation. This review aims to identify potential targets (food patterns, single foods, or individual nutrients) for preventing CVD and quantifies the magnitude of the beneficial effects observed. On the other hand, we analyze the possible mechanisms implicated in this cardioprotective effect.

## 1. Introduction

Data obtained in 2013 showed that the leading global cause of death in Western countries is cardiovascular disease (CVD), accounting for 17.3 million of all deaths worldwide per year (or 31.5% of all global deaths), despite steadily decreasing during the past 10 years [1,2]. One in three deaths in the United States and one in four deaths in Europe are caused by CVD [3]. So, in 2035, 45.1% (>130 million adults) of the US population are projected to have clinical expression of CVD [1,4]. CVD describes a range of disorders that affect the heart and blood vessels, such as hypertension, stroke, atherosclerosis, peripheral artery disease, and vein diseases [4]. The probability of developing CVD is associated with unhealthy dietary patterns (i.e., excessive intake of sodium and processed foods; added sugars; unhealthy fats; low intake of fruit and vegetables, whole grains, fiber, legumes, fish, and nuts), together with a lack of exercise, overweight and obesity, stress, alcohol consumption, or a smoking habit (Figure 1) [5,6,7]. Additionally, CVD often coincides with multiple co-morbidities, such as obesity, diabetes, hypertension, or dyslipidemia, which represent four of the 10 greatest risk factors for all-cause mortality worldwide [8]. Furthermore, the rising incidence of CVD over the last 25 years has become a public health priority, especially the prevention of CVD (or cardiovascular events) through lifestyle interventions [9]. On the one hand, a large body of scientific evidence has reported that nutrition might be the most preventive factor of CVD death [10], and could even reverse heart disease [8]. On the other hand, diet seems to play an important role in the management of other risk factors, such as excess weight, hypertension, diabetes, or dyslipidemia [8]. In this sense, the identification and classification of nutrients, foods, or dietary patterns that can enhance CVD prevention is a priority.

Atherosclerosis is an inflammatory disease that contributes to major incidence and mortality of CVD. Oxidative stress and systemic inflammation are modifiable by nutrition [10,11,12,13], with an excess energy intake and physical inactivity as contributors of pro-inflammatory cytokines’ secretion [14]. Inflammatory processes involve the sub-endothelial area of the arterial wall, accumulating lipids and lipid-laden macrophages among other cell types [15,16]. Current scientific evidence shows that chronic inflammation plays a key role in the pathogenesis of coronary artery disease (CAD), including the initiation and progression of atheroma plaque and rupture, and post-angioplasty and restenosis [17]. The main mediators of CAD development are C-reactive protein (CRP), interleukin (IL)-1, IL-6, IL-8, IL-1β, IL-18, monocyte chemoattractant protein (MCP)-1, and tumor necrosis factor (TNF)-α, among others. Moreover, those mediators are considered potential inflammation biomarkers and their expression may correlate with CAD severity [17,18,19].

As such, current evidence indicates that Western dietary patterns compared to healthier dietary patterns, such as the “Mediterranean diet” (MeDiet), leading to an excessive production of proinflammatory cytokines associated with a reduction of the synthesis of anti-inflammatory cytokines [20,21,22,23]. Therefore, the intake of fruits, vegetables, whole grains, nuts, seeds, and legumes is associated with lower inflammation [24,25,26,27,28], whereas red meat intake has been correlated with a higher inflammatory level [24,29,30,31]. Consequently, increased adherence to healthier dietary patterns, characterized by higher intake of fruits, vegetables, legumes, nuts, and whole grains, may mitigate low grade inflammation, preventing CVD [32,33,34,35].

In addition, microbiota has been linked to intestinal health, the immune, system and bioactivation and metabolism of nutrients, such as vitamins B and K and bioactive compounds. Recent clinical studies suggest a correlation between elevated plasma trimethylamine N-oxide (TMAO), which is produced by gut bacteria metabolism of dietary components, such as L-carnitine, betaine, and choline, and a higher risk of diabetes, hypertension, and atherosclerosis [36,37,38]. Therefore, it has been well studied that diet affects the composition and activity of gut microbiota and situations of gut microbiota dysbiosis may be involved in the development of CVD.

This review aims to identify potential targets (food patterns, single foods, or individual nutrients) for CVD prevention, quantify the magnitude of the beneficial effects observed, and analyze the mechanisms implicated in these cardioprotective effects. Besides, studies were limited to humans with no time restriction. Relevant studies, systematic reviews, and meta-analyses were searched to obtain the reference lists. The Medical Subject Headings search terms included: Inflammation, oxidative stress, inflammatory markers, IL, CRP, TNF-α, IL-6, dietary pattern, Mediterranean diet, Dietary Approach to Stop Hypertension (DASH diet), atherosclerosis, fruits and vegetables, olive oil, nuts, wine, fiber, micronutrients, vitamins, minerals, omega-3 fatty acids, lycopene, phytosterols, and polyphenols.

## 2. Atherosclerosis

Early stages of atherosclerosis are involved the internalization of lipids in the intima, mainly low-density lipoproteins (LDL), which is translated to endothelial dysfunction [39]. The disruption of the endothelial function promotes the inflammatory response, thrombus formation, and multiple pathological consequences, such as calcifications, stenosis, rupture, or hemorrhage [15,40].

The inflammatory response is enhanced by the infiltration of low-density lipoproteins (LDL) particles in the extracellular matrix (EM) while circulating monocytes are attached to the endothelium and transformed into macrophages infiltrating into the sub-endothelial area. The retention of LDL in EM is mediated by proteoglycans, which facilitate retention in the intima [41]. The LDL particles attached in the intima are susceptible to oxidative modifications by reactive oxygen species (ROS) and enzymatic modification released from inflammatory cells. Macrophages are converted into foam cells after oxidized LDL (oxLDL) particles are absorbed by them. Additionally, endothelial dysfunction enhances platelets’ adhesion, which secretes chemotactic substances and growth factors, promoting plaque progression [42]. Vascular smooth muscle cells (VSMC) are also involved in plaque progression. Foam cells’ growth factors and cytokines’ secretion enhance VSMC migration to the intima where they contribute to the formation of the fibrous cap [43]. If the progression of lipid accumulation persists, foam cells and macrophages’ apoptosis is induced jointly with pro-thrombotic molecules’ secretion [44,45]. Atherosclerotic plaque progression and plaque disruption, promoted by pro-thrombotic agents, initiate platelet activation and aggregation, which leads to the coagulation cascade and, consequently, thrombus formation [46]. The clinical manifestations of advanced atherosclerosis are coronary heart disease, ischemic stroke, peripheral artery disease, heart failure, or sudden death [47].

## 3. Oxidative Stress and Inflammation

Oxidative stress has been related to the pathogenesis of atherosclerosis [48]. ROS and reactive nitrogen species (RNS) are mainly produced through mitochondrial activity and other pathways, such as nitric oxide (NO) synthase, and oxidase enzymes, such as Nicotinamide adenine dinucleotide phosphate (NADPH) oxidases (Nox), xanthine oxidase (XO), lipoxygenase, myeloperoxidase, uncoupled endothelial nitric oxide synthase (eNOS), and the mitochondrial respiratory chain via a one-electron reduction of molecular oxygen. Note the role of Nox in oxidative stress, as upregulated and overactive Nox enzymes contribute to oxidative stress and CVD. Several signaling pathways regulate inactivation and degradation of ROS and RNS, including catalase, glutathione peroxidase, and superoxide dismutase among others. An excess of ROS and RNS leads to oxidative stress, promoting cell proliferation, migration, autophagy, necrosis, DNA damage, endoplasmic reticulum stress, endothelial dysfunction, and higher levels of oxLDL [49,50]. Moreover, ROS activate the inflammatory response that directly affects plaque progression and endothelial function, increasing the levels of inflammatory cytokines, such as interleukins (IL-6, IL-8), TNF-α, and MCP-1, and adhesion molecules, such as intercellular adhesion molecule 1 (sICAM-1) and vascular cell adhesion molecule (sVCAM-1) [51]. Simultaneously, transcription factors activation, mainly nuclear factor kappa B (NF-κβ) and nuclear factor (erythroid-derived 2)-like 2 (Nrf2), and signal transduction cascades result in a high production of inflammatory cytokines and inducible nitric oxide synthase [52]. NO has important anti-inflammatory, antihypertensive, and antithrombotic actions due to its strong vasodilator activity and anti-platelet aggregation. Additionally, anti-inflammatory effects are enhanced by the ability of NO to inhibit NF-κβ expression and the subsequent adhesion molecules [53]. Oxidative stress contributes to endothelial eNOS dysfunction [54,55]. Dysfunctional eNOS generates superoxide anions instead of NO, which is translated to a higher ROS production and contributes to atherogenesis [56]. In the case of inducible NOS (iNOS), which is expressed in cells after cytokines or bacterial lipopolysaccharide stimulation, an excessive and sustained production of NO has been linked with inflammatory diseases and septic shock [57]. Therefore, a decrease of NO production by eNOS leads to endothelial dysfunction while an excessive NO production by iNOS may induce pro-inflammatory and pro-atherogenic factors.

The causes and risk factor of atherosclerosis and oxidative stress are not well defined. However, certain health conditions and habits may contribute to atherosclerosis development, such as high total cholesterol and low-high-density lipoprotein cholesterol (HDL-c) levels, hypertension, type 2 diabetes mellitus (T2DM), obesity, and physical inactivity. Additionally, healthy dietary patterns and lifestyle modifications are potential strategies for atherosclerosis and oxidative stress prevention.

## 4. Dietary Patterns

Several studies correlate healthy dietary patterns with lower plasmatic concentrations of pro-inflammatory markers [58], whilst a Western-type diet (meat-based dietary pattern) is associated with higher levels of low-grade inflammation [31]. For that reason, CVD guidelines recommend a healthy diet. Dietary intervention allows a better combination of multiple foods and nutrients. Therefore, healthy dietary patterns support a greater magnitude of beneficial effects than the potential effects of a single nutrient supplementation, because of the synergistic health effects among them. The current body of evidence shows that healthy dietary patterns share similarities, such as a high intake of fiber, antioxidants, vitamins, minerals, polyphenols, monounsaturated, and polyunsaturated fatty acids (MUFA and PUFA, respectively); low intake of salt, refined sugar, saturated, and trans fats; and carbohydrates of low glycemic load [59]. This translates to a high intake of fruits, vegetables, legumes, fish and seafood, nuts, seeds, whole grains, vegetable oils (mainly, extra virgin olive oil [EVOO]), and dairy foods together with a low intake of pastries, soft drinks, and red and processed meat [59,60].

Mediterranean and DASH dietary interventions are well studied for CV outcomes. Both dietary patterns may reduce the incidence CVD through the down-regulation of low-grade inflammation and better control of body weight, which also improve other risk factors, and are correlated with lower numbers of clinical events [59,60]. Thus, this will be the focus of this research.

### 4.1. Mediterranean Diet

Until now, the main benefits of the Mediterranean diet (MeDiet) against CVD (Figure 2) have been associated with a better control of risk factors to improve blood pressure (BP), lipid profile, glucose metabolism, arrhythmic risk, or gut microbiome [59]. Also, some authors have suggested that MeDiet may exert an anti-inflammatory effect (Table 1) in the vascular wall as a possible mechanism to explain the link between MeDiet and low CVD prevalence [61]. Interestingly, MeDiet seems to modulate the expression of pro-atherogenic genes as cyclooxygenase-2 (COX-2), MCP-1, and low-density lipoprotein receptor-related protein (LRP1) [62], reducing plasmatic levels of plaque stability and rupture related molecules as MMP-9, IL-10, IL-13, or IL-18 [63,64].

The observational study, ATTICA, evaluated the link between MeDiet and the incidence of metabolic syndrome (MetS) in 1514 men and 1528 women (>18 y) without clinical evidence of CVD or any other chronic disease [65] during 10 years. Authors found that an increase of 10% in the MeDiet adherence score was associated with a 15% lower odds for CVD incidence. Nevertheless, the inflammatory factors studied (adiposity, CRP, IL-6), whose components are associated with a higher likelihood of CVD, showed a higher incidence (29%) in those subjects away from the MD [65]. Also, the Multi-Ethnic Study of Atherosclerosis (MESA) investigated if a dietary quality score based in a MeDiet pattern was related with regional adiposity [66]. Authors studied 5079 individuals free of CVD (61 ± 10 years) and found that a high quality dietary pattern was associated with less regional adiposity and a lower body mass index (BMI), CRP, and insulin resistance. Thus, Lahoz et al. [67] conducted a cross-sectional analysis of 1411 subjects of the Screening PRE-diabetes and type 2 DIAbetes (SPREDIA-2) study (mean age 61 years, 43.0% males) to assess whether the 14-point Mediterranean Diet Adherence Screener (MEDAS) was associated with serum CRP levels. After adjusting for confounders, the authors showed an inverse correlation between the adherence to MeDiet and CRP levels (*p* = 0.041). Also, a substudy of the MOLI-SANI cohort (6879 women and 6892 men) found that men with a higher adherence to a healthy high-antioxidant diet (HAC), vitamins, and phytochemicals enriched diet, inside a MeDiet pattern, were more protected against hypertension and inflammation than those with a healthy low-antioxidant diet [68]. Authors found HAC was associated with a significant decrease in CRP levels (β = 0.03, *p* = 0.03). Finally, Sureda et al. [69] conducted a study of two cross-sectional nutritional surveys with men and women (219 males and 379 females) aged among 12–65 years old, who lived on the Balearic Islands. Results showed that the male adult population with a higher adherence to the MeDiet showed lower concentrations of pro-inflammatory biomarkers, such as TNF-α and hs-CRP. Also, in this population, lower levels of leptin or plasminogen activator inhibitor 1 (PAI-1) were observed, while adiponectine levels were increased. Moreover, females (young and old), with a higher adherence to the MeDiet, showed lower hs-CRP levels. Lower leptin levels were showed only in the young female group, while PAI-1 reduction was only observed in female adults.

On the other hand, the PREDIMED (Prevención con Dieta Mediterránea) study, the largest interventional study about MeDiet, which included 7447 subjects (55 to 80 years of age, 57% women) at high CV risk, without CVD at baseline, showed a lower prevalence of CV events in participants assigned to a MeDiet supplemented with extra-virgin olive oil (EVOO) or nuts than those assigned to a low-fat diet after five years intervention [70]. Focusing on the PREDIMED study, the MeDiet has reported an anti-inflammatory effect on the expression of adhesion molecules in leukocytes, but also improvements in the circulating levels of soluble adhesion molecules (sVCAM-1, sICAM-1, E- and P-selectin, cytokines (IL-1, IL-6, IL-8, IL-12p70, CRP, TNF-α, tumor necrosis factor receptor (TNFR)-60 and 80, etc.), chemokines (MCP-1, Regulated on Activation, Normal T Cell Expressed and Secreted [RANTES], macrophage inflammatory proteins [MIP-1β], etc.) and molecules related with vulnerability atheroma plaque (IL-10, IL-13, IL-18, or Matrix metallopeptidase-9 [MMP-9]) after three months, one, three, and five years intervention in those participants that followed a MeDiet supplemented with EVOO or nuts [63,64,71,72,73,74]. On the one hand, the results obtained support that MeDiet exerts an important immunomodulatory effect, reducing proinflammatory biomarkers, especially those related to atheroma stability plaque. On the other hand, these anti-inflammatory effects seem to appear in the short and medium-term (three months, one year) and later is maintained in the long-term (three and five years). In a Swedish randomized crossover study [75], the adherence to a Medierranean-style diet and its correlation with inflammatory biomarkers (CRP, IL-6), vasoregulation, vascular endothelial growth factor (VEGF), and serum phospholipid fatty acid composition was investigated. The 22 subjects free of CVD (10 women) received a Mediterranean-type diet or Swedish diet for four weeks. No changes were observed for CRP or IL-6 although the MeDiet group showed significant reductions in leukocytes and platelets levels (by 10% and 15%, respectively) and in VEGF levels (15%). Esposito and et al. [76] assessed the effects of a Mediterranean dietary pattern on endothelial function and vascular inflammatory markers in patients with the MetS during two years (90 patients/intervention group). Compared with controls, the MeDiet group showed lower serum concentrations of high-sensitivity ((hs)-CRP, IL-6, IL-7, and IL-18 (*p* ≤ 0.04; all)) and improved the endothelial function score (*p* < 0.001), defined as a measure of BP and platelet aggregation response to l-arginine. These results are in agreement with Azzini et al. [77], who also reported improvements in the CV risk profile and modulation of inflammatory levels (IL-10 and TNF-α), and a reduction in oxidative stress (malondialdehyde [MDA]).

The influence of polyphenols’ content of a MeDiet pattern was also studied in the PREDIMED study [78]. Total polyphenol excretion (TPE) in urine was analyzed in 1139 participants homogenously and randomly in one of the three groups. Authors observed that MeDiet (supplemented with EVOO or nuts) significantly increased their TPE after one year of dietary intervention, decreasing inflammatory biomarkers compared with baseline (sVCAM-1, sICAM-1, IL-6 MCP-1, TNF-α).

Epigenetic studies have also reported similar results. Thus, Arpón et al. [79,80] conducted a substudy on 36 participants of the PREDIMED cohort after five years of intervention, where the main results were that a MeDiet supplemented with EVOO or nuts may influence the methylation status of peripheral white blood cell (PWBCs) genes. These changes were mainly observed in genes related to intermediate metabolism, diabetes, inflammation, and signal transduction. However, interactions among MeDiet and COX-2, IL-6, apolipoprotein A2 (APOA2), cholesteryl ester transfer protein plasma (CETP), and transcription factor 7-like 2 (TCF7L2) gene polymorphisms have been demonstrated [81,82,83,84].

### 4.2. DASH Diet

A large body of evidence supports that adherence to a DASH dietary pattern is linked to improvements in BP [85], body weight [86], glucose-insulin homeostasis [87], blood lipids and lipoproteins [88], inflammation grade [89,90], endothelial function [91,92], the gut microbiome [93,94], CVD risk [95,96], and total mortality [97,98]. The DASH diet is characterized by a high intake of fruits and vegetables, legumes, low fat dairy, whole grain products, nuts, fish, and poultry; a reduced intake of saturated fat, red meat and processed meats, and sweet beverages; and a low intake of sodium and refined grains [89,99].

Focusing on inflammatory markers and oxidative stress, several studies have shown the protective effect of the DASH diet on CVD (Table 1). A recent systematic review and meta-analysis of randomized trials [89], which included six randomized control trials (RCT) with 451 participants who were followed for 3–24 weeks, studied the effect of the DASH diet on inflammatory biomarkers. Results showed that the DASH diet significantly reduced high hs-CRP concentrations (mean difference (MD) = −1.01, 95% confidence interval (CI): −1.64, −0.38; I-squared (I2) = 67.7%) compared to other diets. When the follow up of participants was longer, the reduction of hs-CRP serum levels was greater. However, when the DASH diet effect was compared to other healthy diets, no significant changes were observed. Besides, a meta-analysis conducted by Neale et al. [100] about 17 RCTs observed that following a healthy diet (MeDiet, Nordic diet, Tibetan diet, and DASH diet) was associated with a significant reduction of CRP levels (−0.75, 95% confidence interval (CI): −1.16, −0.35; *p* = 0.003). No changes were found for the other biomarkers (TNF-α, total adiponectin, high-molecular-weight adiponectin, adiponectin:leptin ratio, resistin, or retinol binding protein 4). Eichelmann et al. [101] studied the link between plant-based diets (Nordic diet, MeDiet, vegetarian diet, plant-based diet, Paleolithic diet, and DASH) and obesity-related pro-inflammatory markers (CRP, IL-6, TNF-α, sICAM-1, leptin, adiponectin, and resistin) on 29 interventional trials with a total of 2689 participants. Results showed improvements in obesity-related inflammatory profiles after following plant-based diets: CRP (−0.55 mg/L), IL-6 (−0.25 ng/L), and sICAM-1 (−25.07 ng/mL). No significant changes were observed for TNF-α, resistin, adiponectin, and leptin.

In a cross-sectional analysis of 1493 men and women (aged 50–69 years), potential associations between dietary quality (through the DASH dietary quality score), adiposity, and biomarkers of glucose metabolism, lipid profile, and inflammation were assessed [102]. Results showed that a higher adherence to the DASH dietary pattern was associated with improvements in adiposity measures (BMI, *p* < 0.05, waist circumference, *p* < 0.001), and lower concentrations of TNF-α, IL-6, CRP, WBC, and PAI-1 (*p* < 0.05; all), such as pro-inflammatory, pro-thrombotic, and pro-atherogenic markers. Improvements in lipoprotein profile parameters (LDL-c, HDL-c, and lower large very low density lipoprotein [VLDL] particles, *p* < 0.001 all) and glucose homeostasis biomarkers (HOMA, insulin and glucose, *p* < 0.05 all) were also shown.

With respect to interventional studies, the DASH diet has reported improvements on insulin resistance, inflammation, and oxidative stress in women with gestational diabetes [103]. The RCT was performed with 32 pregnant women diagnosed with gestational diabetes (GD) at 24- to 28-week gestation. All of them were randomly assigned to a DASH diet group or a control group (16 participants/group) and were followed up for four weeks. The DASH diet compared with the control showed significant reductions in serum insulin levels (−2.62 μIU/mL, *p* = 0.03), fasting plasma glucose (FPG) (−7.62 mg/dL, *p* = 0.02), and the homeostasis model of assessment-insulin resistance (HOMA-IR) score (−0.8, *p* = 0.03). Furthermore, Kawamura et al. [104] also reported significant reductions in the BMI, BP, FPG, and fasting insulin level (*p* ≤ 0.003; all) after analyzing 58 Japanese participants with untreated high-normal BP or stage 1 hypertension (30 men and 28 women; mean age 54.1 ± 8.1 years), who followed a modified DASH diet (salt 8.0 g/day) during two months. Finally, the DASH diet also increased the plasma total antioxidant capacity (45.2 mmol/L, *p* < 0.0001) and total glutathione levels (108.1 μmol/L, *p* < 0.0001). Neither group showed changes in serum hs-CRP levels. Saneei et al. [105] conducted a cross-over study, which examined the effects of the DASH diet on markers of systemic inflammation in 60 post-pubescent girls with MetS (aged 11–18 years and weight mean was 69 kg). Participants were randomized into two groups: The DASH diet or usual dietary advice (control group) and were followed for six weeks. Results did not show significant changes on TNF-α, IL-2, IL-6, and adiponectin levels, whereas hs-CRP levels were significantly lower (−0.09 mg/L, *p* = 0.002) in those participants with higher adherence to a DASH diet compared to the control group.

## 5. Foods

### 5.1. Fruits and Vegetables

The European Society of Cardiology (ESC) and American Heart Association Nutrition Committee strongly endorse the daily consumption of multiple servings of both fruits and vegetables in order to reduce CVD risk [106,107]. These recommendations are based upon epidemiological studies and meta-analysis, mainly [106,107,108,109,110,111,112,113]. A recent meta-analysis [108] with 83 studies (71 clinical trials and 12 observational studies) showed that a higher intake of fruit or vegetable was significantly inversely associated with CRP and TNF-α levels (*p* < 0.05; both) and directly associated with an increased proliferation of γδ-T cell populations (*p* < 0.05). Also, Corley et al. [111] studied, in 792 participants aged 70 years from the Lothian Birth Cohort 1936, the association between biomarkers of systemic inflammation (such as CRP and fibrinogen) and specific single foods (fruits and vegetables). The dietary intake was measured using a 168-item Food Frequency Questionnaire (FFQ). Authors described that a higher fresh fruit intake was associated with lower CRP levels (≤3 mg/L) (β = 0.100, 95% CI 0.82, 0.99). No significant association was found between vegetables and CRP. Similar results (*p* < 0.05) were found between fibrinogen levels and fruit intake (β = 0.083) or combined fruits and vegetables intake (β = 0.084).

Also, in the cross-sectional study conducted by Holt et al. [112], in 285 healthy adolescent boys and girls aged 13 to 17 years, it was found that serum CRP levels were inversely associated with fruit intake (r = −0.19; *p* = 0.004), while IL-6 was inversely associated with fruit and vegetable intake and TNF-α only with vegetable consumption (*p* < 0.05; both). The HELENA Cross-Sectional Study [113], which aimed to demonstrate that a healthy diet might reduce adiposity and systemic inflammation, found that fruits and nuts were negatively linked with IL-4 (all subjects, *p* < 0.05; both) and TNF-α (only girls, *p* = 0.036). Contrastingly, vegetables showed only significant inverse correlations with sE-selectin (all subjects, *p* ≤ 0.0012; both). This study was carried out in 464 adolescents (13–17 years) of the European HELENA cohort. In a cross-sectional analysis [114], in 1005 Chinese women aged 40 to 70 years, the association between vegetable intake and inflammatory and oxidative stress markers was studied. Results showed that a higher intake of cruciferous vegetables was associated with lower concentrations of TNF-α (*p* trend = 0.001), IL-1β (*p* trend = 0.004), and IL-6 (*p* trend = 0.02). Additionally, the mean difference of concentrations among the highest and the lowest quintiles of cruciferous vegetables intake were 12.66% for TNF-α, 18.18% for IL-1β, and 24.68% for IL-6. Any association was observed between the consumption of cruciferous vegetable and oxidative stress markers (F2-isoprostanes and 2,3-dinor-5,6-dihydro-15-F2t-IsoP).

Finally, in a sub-study from the PREDIMED study, Urpí-Sardà et al. [77] found that the participants who increased more than 62.7 g/day of their consumption of vegetables after one year decreased their plasma concentration of TNFR60 from 1.7 µg/L to 1.5 µg/L (*p* < 0.05), as shown in Table 2.

### 5.2. Olive Oil

Several studies and meta-analyses have demonstrated the anti-inflammatory effects of olive oil (OO) rich diets [115,116,117]. Bioactive components of EVOO, the main key food of the MeDiet, have demonstrated improvements in inflammatory status, oxidative stress, and endothelial dysfunction [115]. A recent meta-analysis conducted by Schwingshackl et al. [118] in 30 RCT (3106 participants and daily consumption of 1 mg and 50 mg OO) found a significant decrease in CRP (−0.64 mg/L, *p* < 0.0001, n = 15 trials) and IL-6 (−0.29 mg/L, *p* < 0.04, n = 7 trials) compared to controls. Also, the flow-mediated dilatation (FMD) value was significantly increased in subjects with highest OO intake (0.6%, *p* < 0.002).

Several studies from the PREDIMED study and others related to the MeDiet have reported that a MeDiet supplemented with EVOO leads to decreased levels of N-terminal pro-brain natriuretic peptide (NT-proBNP) [119], the progression of intima media thickness (IMT) in those with elevated baseline IMT [120,121], improved systolic and diastolic BP in both hypertensive and non-hypertensive patients [122,123,124], and decreased expression and concentration of circulating inflammatory biomarkers related to atherosclerosis [63,64,71,72,73,74]. Urpí-Sardà et al. [72] reported that those participants with a higher adherence to a MeDiet + EVOO and whose intake of VOO was higher than 24 g/day showed lower plasma concentrations of TNFR60 after one year of intervention. Moreover, Camargo et al. [125] observed that, after isolation of peripheral blood mononuclear cells (PBMCs), the MeDiet exerted an inhibitory effect on the expression of genes related to plaque progression, such as MMP-9, NF-κβ or MCP-1, by increasing IκB kinase (IκBα) expression after the intake of a MeDiet + EVOO compared with two other diets (*p* < 0.05; all). IκBα stabilizes the NF-κβ molecule in the cytoplasm, maintaining it in an “unactivated” state. Widmer et al. [126] observed that daily consumption of 30 mL of OO or 30 mL of OO supplemented with epigallocatechin 3-gallate (EGCG) in 82 subjects (≥18-y) with early atherosclerosis showed improvements of endothelial dysfunction, independently of EGCG supplementation, after four months of intervention. However, the OO + EGCG group showed significant reductions in inflammatory parameters, such as sICAM-1 (*p* ≤ 0.001), white blood cells (*p* < 0.05), monocytes, and lymphocytes (*p* < 0.05; both). Additionally, Oliveras-López et al. [127] showed an increase in plasma antioxidant capacity, antioxidant enzymes-catalase, and glutathione peroxidase, as well as an improvement in the superoxide dismutase (SOD) expression after analyzing 45 healthy adults (age: 21–45 years old, mean BMI: 21.4 ± 0.5 kg/m^2^) who ingested 50 mL of EVOO for 30 days. Along with the other studies, the intake of EVOO seems to have positive effects on endothelial function. Finally, in the VOLOS (Virgin Olive Oil Study), participants with mild dyslipemia were randomized in two groups of intervention (daily 40 mL of EVOO with 166 mg/L of hydroxytyrosol vs. refined OO with 2 mg/L of hydroxytyrosol) during seven weeks. Results showed significant reduction of thromboxane B2 (TXB2) levels of 20% in the EVOO groups [128], as shown in Table 2.

### 5.3. Nuts

Nuts, specifically peanuts and walnuts, have been demonstrated to reduce the CVD morbidity and mortality in numerous large prospective cohort studies [129,130]. Mente et al. [131] predicted that nut intake might offer a preventative risk reduction on heart disease (RR = 0.67 [95% CI: 0.57–0.77]). Also, nut intake is associated with weight loss improvements [132], lower LDL-c levels [71,133,134], hypertension risk [133,135], and T2DM, improving hyperglycemia and insulin resistance [136], and inflammatory and oxidants mediators [134,137,138,139].

In a recent extensive meta-analysis, 23 RCTs were evaluated [139] to investigate the effects of nut intake over some inflammatory biomarkers (CRP, sICAM-1, sVCAM-1, IL-6, E-selectin, TNF-α). Authors found significant reductions of sICAM-1 (−0.17 ng/mL, *p* = 0.01) after nut intake. No changes were observed among the others inflammatory markers. Similar results were found in the meta-analysis conducted by Neale et al. [138]. No significant differences were shown after analyzing a wide number of inflammatory biomarkers, such as CRP, adiponectin, IL-6, sICAM-1, sVCAM-1, and FMD, in a total of 32 RCT studies. A significant improvement in FMD after nut intake was observed.

On the one hand, Yu et al. [137] conducted a cross-sectional study to investigate if nut intake was correlated to inflammatory biomarkers levels (CRP, IL-6, and tumor necrosis factor receptor 2 (TNFR2)) from 5013 participants in the Nurses’ Health Study (NHS) and Health Professionals Follow-Up Study (HPFS) who were non-diabetic. Results showed that a higher nut intake showed lower inflammatory biomarkers levels (CRP: RR = 0.80 [95% CI: 0.69, 0.90, *p*-trend = 0.0003]; and IL-6: RR = 0.86 [95% CI: 0.77, 0.97, *p*-trend = 0.006]).

However, a randomized, parallel-group study on 50 patients with MetS and supplemented with 30 g/day of raw nuts (15 g walnuts, 7.5 g almonds, and 7.5 g hazelnuts) showed significant reductions of plasma concentrations of IL-6 (−1.1 ng/L; *p* = 0.035) compared with a control diet [140]. These results are according with the data showed by Hernández-Alonso et al. [141], who reported, in a randomized cross-over study in 54 participants, significant reductions of fibrinogen, oxLDL, and platelet factor 4 levels in the pistachio-supplemented group compared with control group (*p* < 0.05; all) after four months of intervention. Additionally, the pistachio-supplemented group showed lower IL-6 (−9%) and resistin gene expression (−6%) (*p* < 0.05; both). Similar results (−10.3% for IL-6 and CRP, −15.7% for TNF-α) were showed in a randomized, crossover study for 12-weeks carried out by Liu et al. [142] in 20 Chinese subjects with T2DM with mild hyperlipidemia (nine men and 11 women, mean age of 58 years, and BMI of 26 kg/m^2^). See Table 2.

### 5.4. Wine and Other Fermented Alcoholic Beverages

Nowadays, there is enough evidence from both epidemiologic studies and RCTs to conclude that regular moderate consumption of fermented alcoholic beverages, mainly red wine and beer, has cardioprotective effects and can exert a positive effect on CV risk factors [143,144,145].

#### 5.4.1. Wine

Wine or wine-derived phenolic compounds that exert effects through mechanisms on atherosclerosis are clearly identified. On the one hand, wine and their phenolic compounds decrease oxidation of LDL-c and oxidation stress, and increase in NO, improving endothelial function. Also, ethanol increases HDL-c levels and inhibits platelet aggregation, promotes fibrinolysis, and reduces systemic inflammation [144,146].

Janssen et al. [147] investigated the relationship of wine consumption and CV risk markers (CRP, fibrinogen, factor VII, and PAI-1) in a multi-ethnic sample of 2900 healthy women of middle-age, who were followed up for seven years. Authors concluded that moderate wine consumption may protect against CVD, after observing lower concentrations of CRP (*p* < 0.001), fibrinogen (*p* < 0.001), factor VII (*p* < 0.01), and PAI-1 (*p* < 0.05) compared to abstainers or women that drink little wine.

On the other side, Estruch et al. [148] reported that both red wine and gin have anti-inflammatory properties in the atherosclerotic process through the reduction of fibrinogen levels (−9%) and IL-1α (−21%), as well as lower plasma hs-CRP (−21%), sVCAM-1 (−17%), and sICAM-1 (−9%) levels. Moreover, monocyte expression was significantly reduced by 27–32% (LFA-1, MAC-1, VLA-4). In another randomized, crossover consumption trial in 67 males at high risk of CVD, Chiva-Blanch et al. [149] investigated the effects of ethanol and phenolic compounds of 30 g alcohol/day of red wine, and the equivalent amount of dealcoholized red wine and gin (30 g alcohol/d) for four weeks on the expression of inflammatory biomarkers related to atherosclerosis. Thirty g/day of alcohol of red wine showed an increase in plasma concentrations of IL-10 and decreased macrophage-derived chemokine (MDC). On the other hand, sICAM-1, E-selectin, and IL-6 were reduced by phenolic compounds of red wine. Phenolic compounds also inhibited the expression of LFA-1 in T-lymphocytes and MAC-1, and CCR2 expressions in monocytes. Concentrations of CD40 antigen, CD40 ligand IL-16, MCP-1, and sVCAM-1 were downregulated in both groups: Ethanol and phenolic compounds of red wine. A current study conducted by Roth et al. [150] found that aged white wine presents a greater ability to repair and maintain endothelial integrity than gin. In this randomized, controlled, crossover study, 38 high-risk male volunteers (55–80 years), who received 30 g ethanol/day as aged white wine or gin for three weeks, were evaluated. After intervention, T-lymphocytes’ expression of CD31 and CD40 and monocytes expression of CCR2 and CD36 were lower after consumption of aged white wine. Additionally, for aged white wine, a significant reduction was observed in plasma concentrations of IL-8 and IL-18, sICAM-1 and sVCAM-1. Both beverages showed significant reductions in LFA-1, MAC-1, VLA4, CD40, and CD31 expression, as well as lower concentrations of interferon gamma (IFN-γ). Finally, Estruch et al. [151], in a new study, where wine and gin were compared again, found that wine intake significantly decreased plasma SOD activity [8.1 U/gHb (95% CI: 138, 25; *p* = 0.009)] and MDA levels [11.9 nmol/L (95% CI: 21.4, 2.5; *p* = 0.020)] compared to the gin group, as shown in Table 2.

#### 5.4.2. Beer

Among fermented beverages, beer has a moderate polyphenol content that confers grater cardioprotective effects than spirits and distilled beverages [145]. So, De Gaetano et al. [152] described in a consensus document the effects of moderate beer consumption on health and disease, where they concluded that epidemiological studies showed that low-moderate doses of beer intake protect against CV risk and its effect is comparable to that reported for moderate red wine consumption.

In 1999, Wannamethee et al. [153] studied 7735 British men during 17 years and found that regular beer intake was associated with lower total mortality (HR = 0.84 (CI: 0.71–10.01)).

Finally, Chiva-Blanch et al. [154] performed a randomized, crossover controlled clinical trial with 33 subjects to evaluate the effects of three beverages types: A non-phenolic alcoholic beverage, such as gin; beer, which is a moderate phenolic alcoholic beverage; and a non-alcoholic beer, with the same amount of phenolic compounds. Beer and gin showed improvements in the HDL cholesterol levels (around 5%) and adiponectin (around 7%) compared to the non-alcoholic beer intervention. Related to circulating inflammatory biomarkers, IL-1α levels increased (around 24%) and IL-5 levels decreased around 14% after beer and gin intervention. However, non-alcoholic beer showed significant improvements in homocysteine concentration (decreased by around 6%) and improved folic acid levels around 9%. Related to inflammatory biomarkers, non-alcoholic beer intervention showed significant decreases of IL-6r, IL-15, RANTES, and TNF-β levels, as shown in Table 2.

## 6. Nutrients

It is important to focus on the possible benefits of the intake of specific nutrients to avoid possible deficiencies of these nutrients, which can lead to the development of atherosclerotic disease. We have only included information about fiber, some vitamins, and minerals, but no other nutrients—such as carbohydrates, fats, or proteins—which have also been demonstrated to have a certain effect on the risk of developing atherosclerosis.

### 6.1. Fiber

A wide number of studies and scientific publications have reported the health benefits of dietary fiber intake decreasing cholesterol concentrations and BP, while a deficiency of fiber intake is associated with CVD development [155].

On the one hand, several meta-analyses have displayed that a higher dietary fiber intake is linked with a lower relative risk of total all-cause mortality among 16–23% [156,157,158]. On the other hand, several clinical trials have studied the link between diet and inflammation, and more specifically, the impact of dietary fiber. Although, to date, the implicated mechanisms are still unknown, the proposed mechanisms are that dietary fiber decreases the glucose absorption, and down-regulates the expression of oxidative stress related cytokines or the inflammatory response mediated by gut microbiota exposed to fiber [159].

In an observational study of 1958 postmenopausal women (age 50–79 years), dietary fiber consumption was associated with higher levels of inflammatory markers (CRP and IL-6) [160]. Also, in the Women’s Health Initiative Observational Study (13,745 US men and women), a higher fiber intake (24.7 g/day) was associated with lower plasma concentrations of IL-6 and TNFR2 compared with the lowest fiber intake group (7.7 g/day) [161]. Similar results were expressed by Qi et al. [162], who observed that concentrations of CRP and TNFR2 were among 8% to 18% lower in the highest quintile of cereal fiber intake compared to the lowest quintile. Similar results were obtained by Estruch et al. [163] for CRP levels and other inflammatory cytokines (IL-6, sICAM-1, sVCAM-1), whose decrease was inversely related with dietary fiber intake, but not significantly. Additionally, cross-sectional data (1088 participants without T2DM at baseline and aged 40 y–60 y) from the Insulin Resistance Atherosclerosis Study [164] showed that whole grain products’ intake was inversely related to PAI-1 (β = −0.102; SEM = 0.038; *p* = 0.0077) and CRP plasma concentrations (β = −0.102; SEM = 0.048; *p* = 0.0340).

In interventional studies, North et al. [165] studied, in 554 subjects (192 men, 362 women), the associations between dietary fiber and CRP levels, showing significantly lower CRP concentrations (−25–54%) with higher fiber intakes (≥3.3 g/MJ). An interventional cross-over study [166] with a total of 60 participants (50% patients with newly diagnosed T2DM, 50% nondiabetic subjects) received three isoenergetic meals separated by one week intervals: A high-fiber (16.8 g) meal; a high-fat meal; and a low-fiber (4.5 g) meal. Results showed that a high fiber intake showed lower plasma IL-18 concentrations and greater stimulation of adiponectin plasmatic levels. Finally, significant reductions of TNF-α (−3.7 pg/mL; *p* < 0.001), were observed after whole grain product intake in a randomized parallel arm feeding trial in 49 subjects who were overweight or obese, and low fruits, vegetables, and whole grain products’ intake [167]. See Table 3.

### 6.2. Micronutrients

Nowadays, there are great experimental, epidemiological, and clinical evidence showing how micronutrients’ ingestion may protect against CVD [168,169,170]. Micronutrients exert their protective effect through three possible ways: 1. Reducing endothelial cells damage; 2. improving the production of NO; and 3. inhibiting oxidation of LDL-c [168,169,170]. Both in adolescence and in adulthood, pro-inflammatory biomarkers have been associated with dietary antioxidants, such as Zn, Se, and vitamin C and E, whose deficiency leads to a higher CVD risk [171,172,173,174,175]. Also, a meta-analysis [176] suggested that Mg supplementation might significantly reduce serum CRP levels (−1.33 mg/L, 95% CI: −2.63, −0.02) after analyzing eight RCTs. Similar results were showed in another meta-analysis [177], where after stratifying by the baseline plasma CRP values of ≤3 and >3 mg/L, found significant reductions of CRP levels (1.12 mg/L, 95% CI: −2.05, −0.18, *p* = 0.019) for the last subgroup. Finally, a recent meta-analysis [178] of seven studies (all RCTs) showed that vitamin D-supplemented groups had lower levels of TNF-α (*p* = 0.04) compared with control groups. No differences between vitamin D and control groups were observed for CRP, IL-10, or IL-6. Relative to vitamin E, a recent meta-analysis [179] suggested that supplementation with vitamin E might reduce serum CRP levels (−0.62 mg/L, 95% CI = −0.92, −0.31; *p* < 0.001) after analyzing 12 trials with 246 participants in the intervention arms and 249 participants in control arms.

De Oliveira Otto et al. [180] investigated the association between dietary micronutrients (heme iron, nonheme iron, zinc (Zn), magnesium (Mg), β-carotene, vitamin C, and vitamin E) with inflammatory markers (CRP, IL-6, total homocysteine (tHcy), fibrinogen, coronary artery calcium, and common and internal carotid artery-IMT) and subclinical atherosclerosis in 5181 participants free of diabetes and CVD from the Multi-Ethnic Study of Atherosclerosis (aged 45 y–84 y). Authors found that Mg and nonheme iron were inversely associated with tHcy concentrations, whilst vitamin C was positively associated with tHcy concentrations. Besides, CRP levels were positively associated with Zn and heme iron, whereas Mg concentrations showed an inverse association with CCA-IMT. Finally, no significant association was observed between dietary intake of carotene or vitamin E and inflammatory markers. Wang et al. [181] also reported that serum vitamin D levels were negatively associated with IL-6 (r = −0.244, *p* = 0.002) and hs-CRP (r = −0.231, *p* = 0.004) levels after studying 152 acute stroke patients. Furthermore, for vitamin D, several observational studies have reported that reduced vitamin D levels are linked with endothelial dysfunction and higher arterial stiffness [182,183], and this deficiency might be related to foam cell formation and decreased expression of adhesion molecules in endothelial cells [184].

Recently, Tabesh et al. [185] examined the effects of co-supplementation of vitamin D and calcium on inflammatory biomarkers and adipokines in 118 diabetic participants with insufficient vitamin D levels. The placebo-controlled clinical trial after eight weeks with four intervention groups ((1) vitamin D + calcium placebo; (2) calcium + vitamin D placebo; (3) vitamin D + calcium; or (4) vitamin D placebo + calcium placebo) showed that supplementation with calcium and vitamin D decreased leptin (−9 ± 18 ng/mL), IL-6 (−4 ± 1 pg/mL, *p* < 0.001), and TNF-α (−3.4 ± 1.3, *p* < 0.05) concentrations compared with the placebo. Also, Shargorodsky et al. [186] studied the effect of vitamin C (500 mg), vitamin E (200 IU), coenzyme Q10 (60 mg), and selenium (120 µg) on inflammatory markers in the long-term (six months) in participants at high CVD risk. No significant changes were observed for homocysteine, endothelin, aldosterone, and renin in participants who received antioxidant supplementation, while there was a significant decrease in HbA1C and a significant increase in HDL-c. Large and small artery elasticity was also significantly increased after antioxidant supplementation intake. In addition, an RCT conducted by Ellulu et al. [187] in 64 obese patients, who were hypertensive and/or diabetic, reported the potential anti-inflammatory effect of 500 mg of vitamin C, twice daily. Vitamin C might decline hs-CRP (*p* = 0.01), IL-6 (*p* = 0.001), and fasting blood glucose (*p* < 0.01) after eight weeks of treatment. Christen et al. [188] also conducted a randomized, double-blind, placebo-controlled sub-study from the Women’s Antioxidant and Folic Acid Cardiovascular Study. They tested a daily combination of folic acid (2.5 mg), vitamin B6 (50 mg), vitamin B_12_ (1 mg), or placebo on 300 participants (half for each group). After 7.3 years, the supplemented group showed significant reductions in homocysteine concentrations (−18%), whereas no changes were observed in CRP, IL-6, sICAM-1, and fibrinogen levels, as shown in Table 3.

## 7. Bioactive Compounds

Multiple bioactive compounds (omega-3 fatty acids, lycopene, or polyphenols) present in the diet have been associated with beneficial effects on atherosclerosis development. All of them act to reduce levels of LDL-c, improving inflammatory and oxidative stress biomarkers. Next, we analyze those cited above.

### 7.1. Omega-3 Fatty Acids

PUFAs, as Omega-3 fatty acid (Ω-3 PUFA), α-linolenic acid (ALA), eicosapentaenoic acid (EPA), and docosahexaenoic acid (DHA), have been reported as potential anti-atherogenic agents for the atherosclerotic process [189]. Mechanisms, through which they might reduce CV risk, include improvements in the lipid and lipoprotein profile, oxidation, thrombosis, endothelial function, BP, plaque stability, CV mortality, platelet aggregation, modulating concentration or expression of pro-inflammatory markers (adhesion molecules, cytokines, etc.), and immune cells [190,191,192].

In a meta-analysis conducted by Wang et al. [193] of 16 randomized placebo-controlled trials in 901 participants, it was reported that Ω-3 PUFA intake (0.45–4.5 g/day, for 56 days) increased FMD by 2.30% (95% CI: 0.89, 3.72%, *p* = 0.001) compared with the placebo group. Additionally, a meta-analysis of 38 RCTs [194] reported reductions by 20–30% in serum triglycerides levels in healthy participants after daily consumption of ≥4 g of EPA and DHA through either supplementation or consumption of enriched foods.

In an observational study [195], 102 Japanese individuals with acute coronary syndrome were analyzed and were stratified in three groups: ≤50, 51–74, and ≥75 years. It was found that low serum DHA concentrations leads to CVD, with DHA useful as a predictive marker of endothelial dysfunction. Similar results were reported by Kelley et al. [196].

On the one hand, Cawood et al. [197] showed that patients in the Ω-3 PUFA group (1.8 g EPA + DHA/day), followed up for 21 days, had a lower number of foam cells (*p* = 0.0390) and T-lymphocytes (*p* = 0.0097), less inflammation (*p* = 0.0108), and improved stability of atheroma plaque (*p* = 0.0209) after analyzing data obtained from a randomized placebo-controlled trial. On the other hand, patients who received Ω-3 PUFAs showed lower expression of mRNA for MMP-7 (*p* = 0.0055), -9 (*p* = 0.0048), -12 (*p* = 0.0044), and for IL-6 (*p* = 0.0395) and sICAM-1 (*p* = 0.0142). Similar results were reported by Thies et al. [198], where after administering dietary fish oil supplements (1.4 g EPA + DHA/day) in patients with advanced atherosclerotic plaque, less inflammation, inhibition of macrophages and lymphocytes infiltration and an increase in plaque stability was observed. Besides, several RCTs have pointed out that Ω-3 PUFAs might modulate the expression of cell adhesion molecules (sICAM-1, sVCAM-1, or sP-selectin) as well as CRP, IL-1β, IL-6, serum amyloid A (SAA), TNF-receptor concentrations, TNF-α, or PAI-1 levels among others [195,199], as shown in Table 4.

### 7.2. Lycopene

Lycopene is a lipophilic and an unsaturated carotenoid, present in red-colored fruits and vegetables, such as tomatoes, papaya, or watermelons among others. Epidemiological observational and interventional studies [200,201] suggest that lycopene might reduce atherosclerotic risk, particularly in early stages of atherosclerosis, preventing endothelial dysfunction (NO bioavailability and blood flow) and LDL oxidation. Others mechanisms through which lycopene might exert effects is improvement of the metabolic profile (by impairing cholesterol synthesis) and BP, through reductions in arterial stiffness, and modulation of the expression of pro-inflammatory markers and platelet aggregation [202]. Furthermore, dietary lycopene confers CV benefits and significant reduction in CV mortality and major CV events in postmenopausal women free of CVD or cancer [202]. Focusing on the risk of develops atherosclerosis, several studies have pointed out lycopene’s antioxidant power as a possible mechanism to explain its health benefits [203].

Additionally, in a recent meta-analysis [204], dietary interventions supplemented with tomatoes significantly decreased LDL-c (−0.22 mmol/L; *p* = 0.006), IL-6 (−0.25; *p* = 0.03) and improved FMD by 2.53% (*p* = 0.01), while lycopene supplementation reduced SBP (−5.66 mmHg; *p* = 0.002). In another study with 40 participants with heart failure [205] (lycopene intervention, 29.4 mg/day of lycopene vs. control group), CRP levels decreased significantly in the intervention group, but only in women (*p* = 0.04).

Dietary data collected from the National Health and Nutrition Examination Survey (NANHES) 2003–2006 [206] showed significant inverse associations with tHcy and CRP for dietary lycopene intake (*p* < 0.05).

Therefore, Valderas-Martinez et al. [201] investigated the postprandial effects of a single dose of raw tomatoes (RT), tomato sauce (TS), and tomato sauce with refined olive oil (TSOO) on CVD. In this randomized, cross-over, controlled feeding trial in 40 subjects free of CVD, authors found that tomato intake significantly decreased some inflammatory biomarkers levels, such as LFA-1, IL-6, IL-18, MCP-1, and VCAM-1, and increased plasma IL-10 levels. In another interventional study [207], with 80 subjects, 40 early atherosclerosis cases, and 40 control subjects, the authors pointed out that serum carotenoids concentrations are linked with the risk of atherosclerosis development. They observed that serum lutein was negatively associated with IL-6 (*p* < 0.001) and directly associated with IFN-γ (*p* = 0.002). Moreover, zeaxanthin was inversely associated with VCAM-1 (*p* = 0.001) and apolipoprotein E (*p* = 0.022) levels, while lycopene was inversely associated with sVCAM-1 (*p* = 0.011) and LDL (*p* = 0.046). However, in a single-blind, randomized controlled intervention trial [208] with healthy volunteers (94 men and 131 women, aged 40 y–65 y), no changes were observed for inflammatory markers (oxLDL, sICAM-1, and IL-6), insulin resistance, and sensitivity markers after 12 weeks of dietary intervention, as shown in Table 4.

### 7.3. Phytosterols

A large body of scientific evidence has concluded that a daily dose of 2–3 g of plant sterols or phytosterols is associated with an LDL-c reduction of around 6–15% of the total concentration [209,210]. These reductions were also observed in a meta-analysis conducted by Demonty et al. [211], where after administering a daily dose of 2.15 g of phytosterols, LDL-c was reduced by 8.8%. In fact, plant sterols have been proposed as a complement of statins treatment in order to decrease the risk of CVD. However, available data is inconsistent, so more research is needed. In this other meta-analysis [212] of 20 RCTs with 1308 participants, the effect of phytosterols intake on pro-inflammatory markers was evaluated. Significant reductions of CRP levels (−0.10 mg/dL) were observed after plan sterols’ intake.

In addition, clinical studies have evaluated the association between phytosterols’ consumption and inflammatory markers, such as CRP and cytokines. Although the results showed by Ras et al. [213] are in aggreement with the data reported by Demonty, no changes in CRP levels were observed. In a new study conducted by Ras et al. [209], in 240 hypercholesterolaemic voluntaries who consumed a low-fat spread with added phytosterols (3 g/day) for 12 weeks, no changes in any of the markers evaluated (CRP, SAA, IL-6, IL-8, TNF-α, and sICAM-1) were observed. Devaraj et al. [214] described significant reductions of IL-6 and IL1β levels after the intake of an orange juice-based beverage fortified with sterols (2 g sterols/day). In addition, results of a double-blinded, randomized, crossover trial [215] with 58 hypercholesterolemic participants during 12 weeks of intervention with margarine supplemented with phytosterols (3 g/day) did not show changes in CRP, IL-6, or TNF-α levels, as shown in Table 4.

## 8. Polyphenols

Polyphenols are the most abundant dietary antioxidants present in most plant origin foods and beverages, which possess a wide range of health effects in the prevention of CVD [216]. The most relevant food sources are fruit and vegetables, red wine, black and green tea, coffee, EVOO, and chocolate, as well as nuts, seeds, herbs, and spices [217].

Numerous scientific reports accumulated in the last years suggest that polyphenols might exert their positive effects by delaying progression of atherosclerosis through several mechanisms: Regulation of signaling and transcription pathways, such as NF-κβ; antioxidant systems; prevention of leukocyte migration and later infiltration inside plaque; reduction of adhesion molecules levels; inhibition of the encoding of pro-inflammatory cytokines; reduction of BP because of the enhanced NO production; and improvements of lipid metabolism, coagulation activity, and endothelial function, among others [218,219].

Several epidemiological studies have reported a negative association between consumption of polyphenols or polyphenols-rich foods and CVD [220,221,222]. A meta-analysis [223], including 14 prospective cohort studies with 1,279,804 participants and 36,352 CVD cases, showed that moderate consumption of coffee (three to five cups/day) was associated with a lower CVD risk compared with non-consumers. Similar results were found in the meta-analysis conducted by Larsson et al. [224]. Additionally, in another meta-analysis (14 prospective studies, with 513,804 participants and a median of follow-up of 11.5 years), it was found that a daily intake of ≥3 cups of tea was associated with a lower risk of stroke (−13%) and ischemic stroke (−24%) [225].

Focusing on the action of polyphenols on endothelial function, a large number of studies that have tested red wine, grape juice, black tea, soy, or dark cocoa showed an increase in FMD [226]. Besides, a daily consumption of dark chocolate (50 g) was associated with an improvement of FMD by 3.99% in acute and 1.45% in chronic intake and reduced systolic (−5.88 mm Hg) and diastolic BP (−3.30 mm Hg) [227]. Additionally, and contrary to the expected, acute consumption of black tea was associated with an increase of SBP (5.69 mm Hg) and DBP (2.56 mm Hg) while FMD was increased by 3.40%. Contrary to black tea, green tea led to a significant reduction in LDL-c (−0.23 mmol/L) [227]. Furthermore, alcohol and red wine moderate consumption were associated with an increase of FMD after analyzing 801 individual foods with food frequency questionnaires (FFQs) from the 2000 Hoorn Study (women = 399; age 68.5 ± 7.2 years) [228].

So, several meta-analyses of RCTs [229,230] have reported significant reductions of LDL-c, SBP, fasting glucose, BMI, hemoglobin A1c, or TNF-α levels, and significant increments of HDL-c.

Additionally, hs-CRP, IL-1β, and sP-selectin levels were significantly decreased after anthocyanin extract interventions. A larger cross-sectional study (BELSTRESS) with observational data from 1031 healthy men (49 years on average) found that drinking tea might reduce the inflammatory processes underlying CVD, as tea intake leads to lower levels of CRP (*p* = 0.004), SAA (*p* = 0.001), and haptoglobin (*p* = 0.02) [231].

Anti-inflammatory and immune-modulating effects of polyphenols have also been suggested as a potential pathway for polyphenols’ health benefits. Several RCTs showed the relationship between cocoa polyphenols and inflammatory biomarkers related to atherosclerosis disease progression. In this sense, Vazquez-Agell et al. [232] suggested that the acute consumption of 40 g of cocoa with water might inhibit the NF-κβ transcription and down-regulate adhesion molecules’ production (sICAM-1 and sE-selectin). Moreover, Monagas et al. [233], in a cross-interventional trial, showed that cocoa powder intake led to a reduced expression of adhesion molecules on monocyte surfaces (VLA-4, CD40, and CD36, *p* ≤ 0.028; all) and lower serum levels of soluble adhesion molecules (sP-selectin and sICAM-1; both *p* = 0.007) in 42 subjects (mean age 69.7 years) at high risk of CVD and after four weeks of intervention. Additionally, Basu et al. [234] found that green tea consumption, as a beverage or extract, did not alter inflammatory biomarkers (adiponectin, CRP, IL-6, IL-1β, sVCAM-1, sICAM-1, leptin, or leptin:adiponectin ratio) related to MetS after eight weeks of intervention. Only SAA was significantly decreased after green tea beverage and extracts’ intake (*p* < 0.005). Also, Zhang et al. [235] reported a significant decrease of CXCL7 by 12.32%, CXCL5 by 9.95%, CXCL8 by 6.07%, CXCL12 by 8.11%, and CCL2 levels by 11.63% after 320 mg intake of purified anthocyanins during 24 weeks. Similar results were observed after administering 320 mg of purified anthocyanins in 150 hypercholesterolemic patients during 24 weeks [236]. In this case, significant reductions of β-thromboglobulin, sP-selectin, and RANTES (regulated on activation, normal T cell expressed and secreted) were observed. Anthocyanins have also been correlated with lower levels of oxidative stress biomarkers. In an interventional study with 42 overweight and smoker participants, significant reductions in oxLDL and 8-iso-prostaglandin F2α were observed after the supplementation with an extract of maqui berry (162 mg anthocyanins) for 40 days [237]. Isoflavones and stilbens (mainly resveratrol) have also shown anti-inflammatory and immunomodulatory effects. Isoflavones improved CRP concentrations of 117 healthy European postmenopausal women after 50 mg/day of isoflavones intake [238]. Other RCT on postmenopausal American women who received daily 25 g of soy protein supplementation, showed reductions of subclinical atherosclerosis progression by 16% as well as reductions of carotid thickness progression by a mean of 68% [239]. Relative to stilbens, Tomé-Carneiro et al. [240] found significant reductions of oxLDL, apolipoprotein B (ApoB), and LDL-c after analyzing the effect of the intake of a grape supplement containing 8 mg resveratrol for six months. Also, a significant reduction of inflammatory markers levels (IL-18, sICAM-1, and sVCAM-1; *p* ≤ 0.037; all) related with atheroma plaque development was observed in 44 healthy participants who ingested the resveratrol extract during four weeks [241]. Additionally, a randomized, double-blind, placebo-controlled clinical trial conducted by Seyyedebrahimi et al. [242] with 48 diabetics type 2 participants and supplemented with 800 mg/day of resveratrol for eight weeks showed a reduction in plasmatic levels of protein carbonyl content and ROS in PBMCs. Furthermore, after resveratrol supplement intake, an increase in the expression of Nrf2 and SOD was observed, as shown in Table 5.

## 9. Conclusions

We have shown the intimate relationship between nutrition and CVD. Thus, the challenge is in promoting healthy dietary habits as well as an active lifestyle as early as possible in children and young adults. The evidence favors consumption of healthy dietary patterns, such as the Mediterranean diet or DASH diet, against other unhealthy dietary patterns, such as the Western diet, based on a high consumption of salt, added sugars, and saturated and trans-fats. Despite the fact that strong evidence shows the potential health benefits of a great amount of foods, nutrients, bioactive compounds, and dietary antioxidants, such as polyphenols, may exert on CV risk factors or directly on CVD development, it is necessary to conduct more interventional studies with a higher number of cases and longer follow up. To date, a lot of results obtained have produced few conclusions and sometimes, even contradictions. Therefore, due to a lack of information about possible mechanisms implicated in the cardioprotective effect of diet, foods, nutrients, or bioactive compounds, this needs to be more investigated.

## Figures and Tables

**Figure 1 ijms-19-03988-f001:**
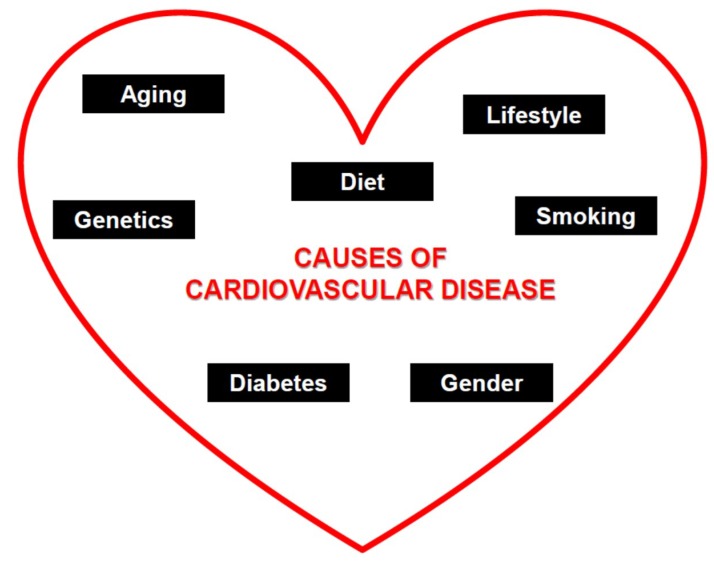
Unhealthy dietary patterns, together with a lack of exercise, overweight and obesity, aging, gender, genetics, or a smoking habit, among others, might lead to the development of cardiovascular disease (CVD).

**Figure 2 ijms-19-03988-f002:**
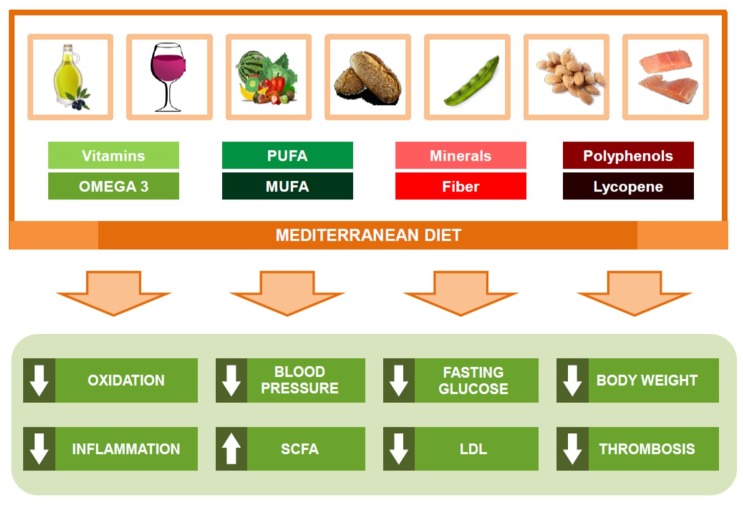
Main protection mechanisms of the Mediterranean diet against cardiovascular disease.

**Table 1 ijms-19-03988-t001:** Potential inflammatory effects of Mediterranean and DASH diet on CVD.

	Pro- and Anti-Inflammatory Markers and Genes	Leukocyte Expression	Oxidative Stress Markers
MeDiet	sVCAM-1, sICAM-1, RANTES, MIP-1β, TNF-α, TNFR-60, IL-1β, IL-6, IL-7, IL-10, IL-12p70, IL-13, IL-18, MMP-9, VEGF, CRP, TCF7L2, APOA2, CETP, COX-2, MCP-1, LRP1	Lymphocytes: CD11a, CD49d, CD40 Monocytes: CD11a, CD11b, CD49d, CD40	MDA, oxLDL
DASH diet	sICAM-1, IL-6, CRP, PAI-1	-	-

APOA2: Apolipoprotein A2; CETP: Cholesteryl ester transfer protein plasma; COX-2: Cyclooxygenase-2; CRP: C-reactive protein; IL: Interleukin; LRP1: Low-density lipoprotein receptor-related protein; MCP-1: Monocyte chemoattractant protein; MDA: Malondialdehyde; MMP-9: Metallopeptidase-9; oxLDL: Oxidized LDL; PAI-1: Plasminogen activator inhibitor 1; sICAM-1: Soluble intercellular adhesion molecule 1; sVCAM-1: Soluble vascular cell adhesion molecule; TNF-α: Tumor necrosis factor; TNFR: Tumor necrosis factor receptor; TCF7L2: Transcription factor 7-like 2; VEGF: Vascular endothelial growth factor.

**Table 2 ijms-19-03988-t002:** Potential inflammatory effects of different foods on CVD.

	Pro- and Anti-Inflammatory Markers and Genes	Leukocyte Expression	Oxidative Stress Markers
Fruits & vegetables	TNF-α, TNFR-60, IL-1β, IL-4, IL-6, γδ-T cell, fibrinogen, sE-selectin	-	F2-isoprostanes, 2,3-dinor-5,6-dihydro-15-F2t-IsoP
Olive oil	sVCAM-1, sICAM-1, RANTES, MIP-1β, TNF-α, TNFR-60, IL-1β, IL-6, IL-7, IL-10, IL-12p70, IL-13, IL-18, MMP-9, CRP, MCP-1, NT-proBNP, NF-κβ	-	plasma antioxidant capacity, antioxidant enzymes-catalase, and glutathione peroxidase
Nuts	CRP, IL-6, TNF-α, TNF-β, TNF-R2, sICAM-1, fibrinogen, PF4, resistin	-	oxLDL
Wine and beer	IL-1α, IL-5, IL-6, IL-6r, IL-8, IL-10, IL-15, IL-18, CRP, MDC, sVCAM-1, sICAM-1, E-selectin, fibrinogen, CD40 ligand, MCP-1, factor VII, PAI-1, IFN-γ, RANTES, TNF-β	Lymphocytes: LFA-1 Monocytes: LFA-1, MAC-1, VLA-4, CCR2, CD36, CD15	SOD, MDA

CCR2: C-C chemokine receptor type 2; CD15: Sialil-Lewis X; CRP: C-reactive protein; IL: Interleukin; LFA: Lymphocyte function-associated antigen 1; MAC-1: T-lymphocytes and macrophage-1 receptor; MCP-1: Monocyte chemoattractant protein; MDA: Malondialdehyde; MDC: Macrophage-derived chemokine; MMP-9: Metallopeptidase-9; NT-proBNP: Pro-brain natriuretic peptide; oxLDL: Oxidized LDL; PAI-1: Plasminogen activator inhibitor 1; PF4: Platelet factor 4; sICAM-1: Soluble intercellular adhesion molecule 1; SOD: Superoxide dismutase; sVCAM-1: Soluble vascular cell adhesion molecule; TNF: Tumor necrosis factor; TNFR: Tumor necrosis factor receptor; IFN-γ: Interferon gamma.

**Table 3 ijms-19-03988-t003:** Potential inflammatory effects of different nutrients on CVD.

	Pro- and Anti-Inflammatory Markers and Genes
Fiber	sVCAM-1, sICAM-1, TNF-α, TNFR2, IL-6, IL-18, CRP, PAI-1
Micronutrients	IL-6, CRP, TNF-α, leptin, tHcy

CRP: C-reactive protein; IL: Interleukin; PAI-1: Plasminogen activator inhibitor 1; sICAM-1: Soluble intercellular adhesion molecule 1; sVCAM-1: Soluble vascular cell adhesion molecule; tHcy: Total homocysteine; TNF-α: Tumor necrosis factor; TNFR: Tumor necrosis factor receptor.

**Table 4 ijms-19-03988-t004:** Potential inflammatory effects of different bioactive compounds on CVD.

	Pro- and Anti-Inflammatory Markers and Genes	Leukocyte Expression
Ω-3 PUFA	sVCAM-1, sICAM-1, sP-selectin, TNF-α, TNFR, IL-1β, IL-6, MMP-7, MMP-9, CRP, PAI-1, SAA	T-lymphocytes
Lycopene	sVCAM-1, IL-6, IL-10, IL-18, MCP-1, tHcy, PAI-1	T-lymphocytes: LFA
Phytosterols	IL-1β, IL-6, CRP	-

CRP: C-reactive protein; IL: Interleukin; LFA: Lymphocyte function-associated antigen 1; MCP-1: Monocyte chemoattractant protein; MMP: Metallopeptidase; PAI-1: Plasminogen activator inhibitor 1; SAA: Serum amyloid A; sICAM-1: Soluble intercellular adhesion molecule 1; sVCAM-1: Soluble vascular cell adhesion molecule; tHcy: Total homocysteine; TNF-α: Tumor necrosis factor; TNFR: Tumor necrosis factor receptor.

**Table 5 ijms-19-03988-t005:** Potential inflammatory effects of polyphenols on CVD.

	Pro- and Anti-Inflammatory Markers and Genes	Leukocyte Expression	Oxidative Stress Markers
Polyphenols	NF-κβ, sICAM-1, sE- and sP-selectin, IL-1β, IL-18, CRP, SAA, CXCL5, CXCL7, CXCL8, CXCL12, CCL2, TNF-α, β-thromboglobulin, RANTES, ApoB	Monocytes: VLA-4, CD40, CD36	oxLDL, 8-iso-prostaglandin F2α, ROS, SOD, Nrf2

ApoB: Apolipoprotein B; CRP: C-reactive protein; CXCL: Chemokine (C-X-C motif) ligand; IL: Interleukin; NF-κβ: Nuclear factor kappa B; Nrf2: Nuclear factor (erythroid-derived 2)-like 2; oxLDL: Oxidized LDL; RANTES: Regulated on activation, normal T cell expressed and secreted; ROS: Reactive oxygen species; SAA: Serum amyloid A; sICAM-1: Soluble intercellular adhesion molecule 1; SOD: Superoxide dismutase; TNF-α: Tumor necrosis factor.

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
