# Peer review of "Nutrition and Cardiovascular Health"

_ijms, 2018, doi:10.3390/ijms19123988_

Reviewer 1 Report

The present manuscripts reviews the dietary impact on cardiovascular diseases of different diets, foods, nutrients and biocompounds. The review is clearly described and focused in the studies demonstrating the relationship between the mentioned parameters. 

I have only one minor comment. It would be interesting to include studies about beer consumption.

Author Response

REVIEWER #1:

The present manuscripts review the dietary impact on cardiovascular diseases of different diets, foods, nutrients and biocompounds. The review is clearly described and focused in the studies demonstrating the relationship between the mentioned parameters.

1. I have only one minor comment. It would be interesting to include studies about beer consumption.

Comment 1: Thank you for this comment. We have included studies about beer as is suggested by Reviewer#1. The new paragraph has been included in lines (453-471) and is highlighted in yellow.

Reviewer 2 Report

This is a review article focused on the relationship between nutrition and cardiovascular diseases. Authors review the results of studies in which the effect of specific diets such as Mediterranean or DASH diet as well as of individual diet components on cardiovascular disease outcomes, oxidative stress markers, inflammatory factors and individual cv risk factors was examined. The topic is of interest and a lot of data are presented. The manuscript is supported by huge literature. However, there are some issues which could be improved.

1)      A lot of data are presented in the paper, however, sometimes they are reported in not well-organized manner. It would be convenient for the reader to include the tables summarizing the results of specific diets/food components on parameters of interest.

2)      Authors address a lot of different aspects in this paper. Indeed, the effect of each separate nutritional intervention such as n-3 PUFA, olive oil, wine, etc. could be the subject for separate review and indeed a lot of such reviews on each of these topics have been published before. It should be clearly specified what are the aims of publishing this paper. The methods of literature search should be described in the introduction

3)      While discussing the results, original studies should not be mixed with meta-analyses. In addition, it would be convenient to separate observational/cross sectional studies from the interventional prospective ones.

4)      Authors present the effect of diets/food constituents on various anthropometric, metabolic, oxidative stress-related and inflammation-related factors, endothelial function and sometimes cv outcomes which makes following the text difficult. It would be suitable to arrange the results of discussed studies according to specific outcome variables, individual cv risk factors and cv morbidity/mortality to make the text more clear

5)      Lines 81-82: the sentence suggests that oxLDL are transformed into foam cells which is not correct; macrophages are transformed into foam cells.

6)      Line 90: “hearth” should be corrected to: “heart”.

7)      Line 98: specific “oxidase enzymes” should be listed. In particular, the role of NOX should be highlighted.

8)      Line 102: “ROS activates” should be corrected to: “ROS activate”.

9)      Line 112: “to” should be inserted between “contributes” and “endothelial”

10)  Lines 100-119: when the role of eNOS and NO is mentioned, it should be highlighted that in contrast to baseline eNOS-derived low quantities of NO, higher NO concentrations produced by iNOS could have the pro-inflammatory and pro-atherogenic roles.

11)  Lines 144-145: it should be specified what “pro-atherogenic genes” and “plaque stability and rupture-related molecules” are mentioned.

12)  Lines 148-149: was prevalence or incidence assessed in this prospective study?

13)  Line 152: BMI is not the inflammatory factor. In addition, which “component 5” is mentioned here?

14)  Lines 151 vs. 153: the discrepancy between 15% and 29% incidence should be clarified.

15)  Line 155: “were more protective” should be corrected to “were more protected”

16)  Lines 175/176: which “endothelial function scores” are mentioned?

17)  Line 186: IL-10 is listed about proinflammatory factors whereas in most systems it exhibits antiinflammatory properties.

18)  Line 277: “inverse associated” should be corrected to “inversely associated”.

19)  Lines 320/321: was the expression of genes (mRNA) measured or only protein concentrations? If mRNA were measured, it should be specified in which cells.

20)  Line 321: IkappaB kinase stimulates NF-kappaB and as such stimulates inflammatory response.

21)  Line 379: decrease in SOD activity does not represent antioxidant effect. Decrease in SOD would be expected to aggravate oxidative stress.

22)  Line 607: increase in blood pressure is not the beneficial effect.

Author Response

REVIEWER #2:

This is a review article focused on the relationship between nutrition and cardiovascular diseases. Authors review the results of studies in which the effect of specific diets such as Mediterranean or DASH diet as well as of individual diet components on cardiovascular disease outcomes, oxidative stress markers, inflammatory factors and individual cv risk factors was examined. The topic is of interest and a lot of data are presented. The manuscript is supported by huge literature. However, there are some issues which could be improved.

1)      A lot of data are presented in the paper, however, sometimes they are reported in not well-organized manner. It would be convenient for the reader to include the tables summarizing the results of specific diets/food components on parameters of interest.

Comment 1: Thank you for your suggestion. We have rewritten all the manuscript and included a new table for each section to summarize the results of specific diet/foods components on parameters of interest. The changes have highlighted in yellow (lines 173-765).

2)      Authors address a lot of different aspects in this paper. Indeed, the effect of each separate nutritional intervention such as n-3 PUFA, olive oil, wine, etc. could be the subject for separate review and indeed a lot of such reviews on each of these topics have been published before. It should be clearly specified what are the aims of publishing this paper. The methods of literature search should be described in the introduction

Comment 2: This is not a systemic review, so we think that this information is not necessary for a Review as ours. However, some terms have been considered and included in this review. The changes are highlighted in yellow (lines 76-81).

“Besides, studies were limited to humans with no time restriction. Relevant studies, systematic reviews and meta-analysis were searched to obtain the reference lists. The Medical Subject Headings search terms included: inflammation, oxidative stress, inflammatory markers, IL, CRP, TNF-α, IL-6, dietary pattern, Mediterranean diet, DASH diet, atherosclerosis, fruits and vegetables, olive oil, nuts, wine, fiber, micronutrients, vitamins, minerals, omega-3 fatty acids, lycopene, phytosterols and polyphenols.”

3)      While discussing the results, original studies should not be mixed with meta-analyses. In addition, it would be convenient to separate observational/cross sectional studies from the interventional prospective ones.

Comment 3: We agree to Reviewer#1, so we have rewritten all the manuscript. The changes are highlighted in yellow (lines 173-765).

4)      Authors present the effect of diets/food constituents on various anthropometric, metabolic, oxidative stress-related and inflammation-related factors, endothelial function and sometimes cv outcomes which makes following the text difficult. It would be suitable to arrange the results of discussed studies according to specific outcome variables, individual cv risk factors and cv morbidity/mortality to make the text more clear.

Comment 4: We agree to Reviewer#1, so we have rewritten all the manuscript. The changes are highlighted in yellow (lines 173-765).

5)      Lines 81-82: the sentence suggests that oxLDL are transformed into foam cells which is not correct; macrophages are transformed into foam cells.

Comment 5: Thank you for this comment. We have changed this sentence. The changes are highlighted in yellow (lines 93-94).

“Macrophages are converted into foam cells after oxidized LDL (oxLDL) particles are absorbed by them”.

6)      Line 90: “hearth” should be corrected to: “heart”.

Comment 6: Thank you for this comment. We have corrected this word. This change is highlighted in yellow (line 104).

7)      Line 98: specific “oxidase enzymes” should be listed. In particular, the role of NOX should be highlighted.

Comment 7: Thank you for your suggestion. We have listed the specific oxidase enzymes and highlighted the role of NOX in CVD and oxidative stress. The changes are highlighted in yellow (lines 111-115).

“oxidase enzymes such as NADPH oxidases (Nox), xanthine oxidase (XO), lipoxygenase, myeloperoxidase, uncoupled endothelial nitric oxide synthase (eNOS), and the mitochondrial respiratory chain via a one-electron reduction of molecular oxygen. Note the role of Nox in oxidative stress, as upregulated and overactive Nox enzymes contribute to oxidative stress and CVD”.

8)      Line 102: “ROS activates” should be corrected to: “ROS activate”.

Comment 8: We have corrected. The change are highlighted in yellow (line 119).

9)      Line 112: “to” should be inserted between “contributes” and “endothelial”

Comment 9: “To” has been inserted between “contributes” and “endothelial”. This change is highlighted in yellow (line 128).

10)  Lines 100-119: when the role of eNOS and NO is mentioned, it should be highlighted that in contrast to baseline eNOS-derived low quantities of NO, higher NO concentrations produced by iNOS could have the pro-inflammatory and pro-atherogenic roles.

Comment 10: Thank you for your observation. We have included a new text and a new reference to explain this. The changes are highlighted in yellow (lines 130-135).

“In the case of inducible NOS (iNOS), which is expressed in cells after cytokines or bacterial lipopolysaccharide stimulation, an excessive and sustained production of NO has been linked with the inflammatory diseases and septic shock [54]. Therefore, a decrease of NO production by eNOS leads to endothelial dysfunction while an excessive NO production by iNOS may induce pro-inflammatory and pro-atherogenic factors”.

Reference: Guzik  TJ, Korbut R, Adamek-Guzik T. Nitric oxide and superoxide in inflammation and immune regulation. J Physiol Pharmacol, 2003, 54, 469-87.

11)  Lines 144-145: it should be specified what “pro-atherogenic genes” and “plaque stability and rupture-related molecules” are mentioned.

Comment 11: Thank you for your suggestion. We have included this information. The changes are highlighted in yellow (lines 165-168).

“Interestingly, MeDiet seems to modulate the expression of pro-atherogenic genes as cyclooxygenase-2 (COX-2), MCP-1 and low-density lipoprotein receptor-related protein (LRP1) [58] reducing plasmatic levels of plaque stability and rupture related molecules as MMP-9, IL-10, IL-13 or IL-18 [59, 60].”

12)  Lines 148-149: was prevalence or incidence assessed in this prospective study?

Comment 12: Thank you for this comment. It was a mistake. It was incidence. Prevalence has been changed by incidence. This change is highlighted in yellow (line 171).

13)  Line 152: BMI is not the inflammatory factor. In addition, which “component 5” is mentioned here?

Comment 13: We have changed BMI by adiposity; perhaps this term is more precise. Beside, we deleted component 5. This change is highlighted in yellow (line 175).

14)  Lines 151 vs. 153: the discrepancy between 15% and 29% incidence should be clarified.

Comment 14: We have rewritten this sentence because it was confusing. This changes is highlighted in yellow (lines 173-176).

“Authors found that an increase of 10% in the MeDiet adherence score was associated with 15% lower odds for CVD incidence. Nevertheless, the inflammatory factors studied (adiposity, CRP, IL-6), which components are associated with higher likelihood of CVD, was showed an higher incidence (29%) in those subjects away from the MD”.

15)  Line 155: “were more protective” should be corrected to “were more protected”

Comment 15: We have corrected it. This change is highlighted in yellow (line 187).

16)  Lines 175/176: which “endothelial function scores” are mentioned?

Comment 16: Thank you for your observation. We have rewritten this sentence and defined this concept. The changes are highlighted in yellow (lines 225-226).

“defined as a measure of BP and platelet aggregation response to l-arginine.”

17)  Line 186: IL-10 is listed about proinflammatory factors whereas in most systems it exhibits antiinflammatory properties.

Comment 17: Thank you for your observation. We use the concept vulnerability atheroma plaque to indicate that some molecules can have a pro- or anti-inflammatory effect on atheroma plaque. If you consider that it’s better using the concepts of stability or instability atheroma plaque we can rewrite this sentence.

18)  Line 277: “inverse associated” should be corrected to “inversely associated”.

Comment 18: Thank you for your observation. We have corrected inverse associated to inversely associated. This change is highlighted in yellow (line 315).

19)  Lines 320/321: was the expression of genes (mRNA) measured or only protein concentrations? If mRNA were measured, it should be specified in which cells.

Comment 19: Thank you for your suggestion. We have rewritten this sentence. The changes are highlighted in yellow (lines 361-362).

“Camargo et al. [122] observed after isolation of peripheral blood mononuclear cells (PBMCs), that MeDiet might exert”

20)  Line 321: IkappaB kinase stimulates NF-kappaB and as such stimulates inflammatory response.

Comment 20:  We are sorry. We don’t undertand this suggestion. IkB stabilizes the NF-kB molecule in the cytoplasm, keeping it as ‘in-activated state. Perhaps, you want to include it in the manuscript, so we have rewritten this sentence. The changes are highlighted in yellow (lines 364-365).

 21)  Line 379: decrease in SOD activity does not represent antioxidant effect. Decrease in SOD would be expected to aggravate oxidative stress.

Comment 21:  We have rewritten this sentence. The changes are highlighted in yellow (lines 449-451).

“Finally, Estruch et al. [151] in a new study where wine and gin were compared again, found that wine intake significantly decreased plasma SOD activity

22)  Line 607: increase in blood pressure is not the beneficial effect.

Comment 22:  Thank you for your suggestion. We have rewritten this sentence. The changes are highlighted in yellow (lines 706-708).

“Additionally and contrary to expected, acute consumption of black tea was associated with an increase of SBP (5.69 mm Hg), DBP (2.56 mm Hg) while FMD was increased by 3.40%.”

Reviewer 3 Report

The authors have written an extensive review on the association between nutrition and cardiovascular diseases.

The paper may be interesting for a broad spectrum of scientist. I believe that the paper could benefit from adding a table that summarizes major nutrients and their effect on the circulatory system.

Finally, increasing research shows a connection between gut bacteria and their metabolites (which are shaped by our diet) and cardiovascular diseases. Adding a short paragraph on the above-mentioned topic would surely make the reviewe more up to date (for review Nowinski A, Nutrition 2018, Konopelski P, Curr Drug Metab. 2018; Huc T, Pharmacol Res. 2018)

Author Response

REVIEWER #3:

The authors have written an extensive review on the association between nutrition and cardiovascular diseases.

The paper may be interesting for a broad spectrum of scientist.

1. I believe that the paper could benefit from adding a table that summarizes major nutrients and their effect on the circulatory system.

Comment 1: According to Reviewer#3 and 2 we have included a new table that summarizes the results of specific diet/foods components on parameters of interest. The changes are highlighted in yellow (lines 173-765).

2. Finally, increasing research shows a connection between gut bacteria and their metabolites (which are shaped by our diet) and cardiovascular diseases. Adding a short paragraph on the above-mentioned topic would surely make the reviewe more up to date (for review Nowinski A, Nutrition 2018, Konopelski P, Curr Drug Metab. 2018; Huc T, Pharmacol Res. 2018)

Comment 2: According to Reviewer#3, we have included a little paragraph in the Introduction section. The changes are highlighted in yellow (lines 67-73).

“In addition, microbiota has been linked to intestinal health, immune system and bioactivation and metabolism of nutrients, such as vitamins B and K and bioactive compounds. Recent clinical studies suggest a correlation between elevated plasma trimethylamine N-oxide (TMAO), which is originated by gut bacteria metabolism of dietary components such as L-carnitine, betaine and choline, with higher risk of diabetes, hypertension and atherosclerosis [36-38].  Therefore, it has been well studied that diet affects the composition and the activity of gut microbiota and situations of gut microbiota dysbiosis may be involved in the development of CVD”.

Round  2

Reviewer 2 Report

The manuscript has been revised according to the reviewers' comments. I appreciate the revision and authors' detailed responses. I have no further concerns.